# Early Change in the Plasma Levels of Circulating Soluble Immune Checkpoint Proteins in Patients with Unresectable Hepatocellular Carcinoma Treated by Lenvatinib or Transcatheter Arterial Chemoembolization

**DOI:** 10.3390/cancers12082045

**Published:** 2020-07-24

**Authors:** Naoshi Odagiri, Hoang Hai, Le Thi Thanh Thuy, Minh Phuong Dong, Maito Suoh, Kohei Kotani, Atsushi Hagihara, Sawako Uchida-Kobayashi, Akihiro Tamori, Masaru Enomoto, Norifumi Kawada

**Affiliations:** Department of Hepatology, Graduate School of Medicine, Osaka City University, Osaka 545-8585, Japan; m2055463@med.osaka-cu.ac.jp (N.O.); hhai@med.osaka-cu.ac.jp (H.H.); thuylt@med.osaka-cu.ac.jp (L.T.T.T.); dongminhphuong15@gmail.com (M.P.D.); maito55jp@gmail.com (M.S.); kouhei-k@med.osaka-cu.ac.jp (K.K.); hagy@med.osaka-cu.ac.jp (A.H.); sawako@med.osaka-cu.ac.jp (S.U.-K.); atamori@med.osaka-cu.ac.jp (A.T.); kawadanori@med.osaka-cu.ac.jp (N.K.)

**Keywords:** HCC, liver cancer, molecular-targeted agent, TACE, tyrosine kinase inhibitor

## Abstract

Immune checkpoint inhibitors, combined with anti-angiogenic agents or locoregional treatments (e.g., transarterial chemoembolization (TACE)), are expected to become standard-of-care for unresectable hepatocellular carcinoma (HCC). We measured the plasma levels of 16 soluble checkpoint proteins using multiplexed fluorescent bead-based immunoassays in patients with HCC who underwent lenvatinib (*n* = 24) or TACE (*n* = 22) treatment. In lenvatinib-treated patients, plasma levels of sCD27 (soluble cluster of differentiation 27) decreased (*p* = 0.040) and levels of sCD40 (*p* = 0.014) and sTIM-3 (*p* < 0.001) were increased at Week 1, while levels of sCD27 (*p* < 0.001) were increased significantly at Weeks 2 through 4. At Week 1 of TACE, in addition to sCD27 (*p* = 0.028), sCD40 (*p* < 0.001), and sTIM-3 (soluble T-cell immunoglobulin and mucin domain–3) (*p* < 0.001), levels of sHVEM (soluble herpesvirus entry mediator) (*p* = 0.003), sTLR-2 (soluble Toll-like receptor 2) (*p* = 0.009), sCD80 (*p* = 0.036), sCTLA-4 (soluble cytotoxic T-lymphocyte antigen 4) (*p* = 0.005), sGITR (soluble glucocorticoid-induced tumor necrosis factor receptor) (*p* = 0.030), sGITRL (soluble glucocorticoid-induced TNFR-related ligand) (*p* = 0.090), and sPD-L1 (soluble programmed death-ligand 1) (*p* = 0.070) also increased. The fold-changes in soluble checkpoint receptors and their ligands, including sCTLA-4 with sCD80/sCD86 and sPD-1 (soluble programmed cell death domain–1) with sPD-L1 were positively correlated in both the lenvatinib and TACE treatment groups. Our results suggest that there are some limited differences in immunomodulatory effects between anti-angiogenic agents and TACE. Further studies from multicenters may help to identify an effective combination therapy.

## 1. Introduction

Primary liver cancer, of which hepatocellular carcinoma (HCC) is the most common type, is the third most common cause of cancer-related deaths worldwide [1,2]. Patients with early-stage HCC can potentially receive curative treatments, such as surgical resection, transplantation, or ablation, and patients at the intermediate stage can undergo transarterial chemoembolization (TACE), but those at an advanced stage of disease are only likely to benefit from systemic therapies [3,4,5]. Molecular-targeted therapies, the two anti-angiogenic tyrosine kinase inhibitors sorafenib [6] and lenvatinib [7] are used as first-line treatments for patients with advanced-stage HCC. Meanwhile, other anti-angiogenic agents like regorafenib [8], cabozantinib [9], and ramucirumab (if α-fetoprotein > 400 ng/mL) [10] have been licensed as second-line treatments. Immune checkpoint inhibitors, such as anti–PD-1 (programmed cell death domain–1) pembrolizumab [11], nivolumab [12] or nivolumab combined with anti–CTLA-4 (cytotoxic T-lymphocyte antigen 4) (ipilimumab) therapy [13], may also be indicated for sorafenib-refractory patients [14,15,16].

However, therapeutic responses to immunotherapy alone are obtained in a minority of patients with HCC. Current clinical trials are therefore focusing on whether combinations of different types of treatments, including anti-angiogenic agents and TACE, may be promising for the enhancement of the antitumor effects of immune checkpoint inhibitors [17]. For example, the combinations of lenvatinib plus pembrolizumab [18] and bevacizumab plus anti–PD-L1 atezolizumab therapy [19] are currently being tested in clinical investigations. In addition, tremelimumab, an anti–CTLA-4 therapy, was more effective when paired with TACE than as monotherapy [20].

In our previous study [21], sorafenib treatment in patients with unresectable HCC provoked dynamic changes in soluble immune checkpoint protein levels as revealed using multiplexed fluorescent bead-based immunoassays. To date, circulating soluble checkpoint proteins, which are part of a family of full-length receptors produced by messenger RNA expression or by the cleavage of membrane-bound proteins, have been studied extensively in various cancers, but not in HCC [22]. Further, changes in the plasma levels of soluble proteins during the early days of treatment with other anti-angiogenic agents or locoregional treatments (e.g., TACE) have yet to be determined.

We hypothesized that lenvatinib and TACE therapies as well as sorafenib would affect plasma levels of soluble immune checkpoint proteins. An understanding of the effects of these therapies on soluble immune checkpoint proteins may provide insight into their immunomodulative characteristics, which would help to develop more effective combination immunotherapies for HCC. In this study, we measured the concentrations of 16 soluble immune checkpoint proteins (Table A1) in plasma samples obtained over four weeks of treatment from patients with unresectable HCC who underwent lenvatinib or TACE treatment.

## 2. Results

### 2.1. Patient Characteristics

The baseline characteristics of patients with HCC in the lenvatinib (*n* = 24) and TACE (*n* = 22) groups are described in Table 1. In brief, the median age of patients was about 75 years (75 vs. 76 years; *p* = 0.530) and males accounted for the majority (75.0% vs. 68.2%; *p* = 0.746) of the population in both treatment groups, with no significant difference apparent between the two groups. Most patients in the TACE group had Barcelona Clinic Liver Cancer (BCLC) [23] stage A (54.5%) cancer, while all patients had BCLC stage B (54.2%) or stage C (45.8%) cancer in the lenvatinib group (*p* < 0.001). Some patients in the TACE group underwent the therapy as their initial HCC treatment, but none in the lenvatinib group (36.4% vs. 0%; *p* = 0.001). No significant differences were found between the lenvatinib and TACE groups with respect to the etiology of liver disease, Eastern Cooperative Oncology Group performance status score, aspartate and alanine aminotransferase (AST and ALT) levels, gamma-glutamyl transferase level, hepatic reserve (Child–Pugh class or albumin–bilirubin score [24,25]) and α-fetoprotein and des-γ-carboxy prothrombin levels as tumor markers.

### 2.2. Changes in Plasma Soluble Checkpoint Protein Levels at Week 1 after the Initiation of Lenvatinib

First, we investigated whether lenvatinib affects the plasma levels of immune checkpoint proteins in the early phase of treatment. As previously reported [21], we analyzed soluble checkpoint protein levels in patients with HCC at Week 1 of lenvatinib treatment. Ultimately, a significant decrease was observed in the level of soluble cluster of differentiation 27 (sCD27) (*p* = 0.040) and significant increases were found in the levels of sCD40 (*p* = 0.014) and soluble T-cell immunoglobulin and mucin domain–3 (sTIM-3) (*p* < 0.001) when compared with at baseline. Meanwhile, no significant changes were found in soluble herpesvirus entry mediator (sHVEM) and the other 12 immune checkpoint proteins (Figure 1 and Figure A1).

### 2.3. Changes in Plasma Soluble Checkpoint Protein Levels at Weeks 2 through 4 after the Initiation of Lenvatinib

Next, we sought to reveal changes in immune checkpoint protein levels at the later stage of lenvatinib therapy. We analyzed soluble checkpoint protein levels in the plasma of patients with HCC at Weeks 2 through 4 of lenvatinib treatment. The increase in sCD27 level was significant (*p* < 0.001). A trend toward increasing sCD40 and sHVEM levels was also observed, but no change reached statistical significance (*p* = 0.070 and 0.090). Also, the change in sTIM-3 was no longer significant at this time point (Figure 2), while the other 12 immune checkpoint proteins showed similar outcomes (Appendix A
Figure A2).

### 2.4. Changes in Plasma Soluble Checkpoint Protein Levels at Week 1 after TACE

Lenvatinib can suppress tumor blood flow by its pharmacological anti-angiogenic effects. To establish a contrast with those who received lenvatinib, we also investigated patients who underwent conventional TACE, which disrupts tumor blood flow via the artificial embolization of hepatic arteries. We analyzed soluble immune checkpoint protein levels in the plasma of patients with HCC at 1 week after TACE. In addition to sCD27 (*p* = 0.028), sCD40 (*p* < 0.001), sTIM-3 (*p* < 0.001), and sHVEM (*p* = 0.003) levels, which exhibited significant (or marginal) changes at Week 1 or Weeks 2 through 4 of lenvatinib treatment, the levels of another six proteins—namely, soluble Toll-like receptor 2 (sTLR-2) (*p* = 0.009), sCD80 (*p* = 0.036), sCTLA-4 (soluble cytotoxic T-lymphocyte antigen 4) (*p* = 0.005), soluble glucocorticoid-induced TNFR-related protein (sGITR) (*p* = 0.030), soluble glucocorticoid-induced TNFR-related ligand (sGITRL) (*p* = 0.090), and sPD-L1 (soluble programmed death-ligand 1) (*p* = 0.070)—were also increased (Figure 3). However, the levels of the remaining six immune checkpoint proteins showed no significant changes (Appendix A
Figure A3).

### 2.5. Relationships between Fold-Changes in Plasma Soluble Immune Checkpoint Protein Levels

The correlations between the fold-changes in the soluble forms of immune checkpoint proteins in plasma at Week 1 of lenvatinib treatment are shown in Figure 4. The fold-changes in soluble checkpoint receptors and their ligands, including sCTLA-4 with sCD80 (*p* < 0.001; r = 0.82)/sCD86 (*p* < 0.001; r = 0.78), and sPD-1 with sPD-L1 (*p* < 0.001; r = 0.91), were positively correlated (Figure 4a). Among the three soluble checkpoint proteins with significant change at Week 1, sCD40 was positively correlated with some proteins in fold-changes; sCD86 (*p* = 0.005; r = 0.62), sPD-1 (*p* = 0.010; r = 0.58), sPD-L1 (*p* = 0.046; r = 0.48) (Figure 4b), sCD28 (*p* = 0.012; r = 0.57), sTLR-2 (*p* = 0.048; r = 0.46), and sHVEM (*p* = 0.042; r = 0.47).

At Weeks 2 to 4 of lenvatinib treatment, the fold-changes in soluble immune checkpoint receptors and their ligands in plasma, including sCTLA-4 with sCD80 (*p* = 0.055; r = 0.55)/sCD86 (*p* = 0.009; r = 0.71) and sPD-1 with sPD-L1 (*p* < 0.001; r = 0.86), were also positively correlated (Appendix A
Figure A4a). Separately, the fold-change in sCD27 was positively correlated with those of sCD86 (*p* = 0.030; r = 0.61), sPD-1 (*p* = 0.010; r = 0.70), and sPD-L1 (*p* = 0.050; r = 0.56), respectively (Appendix A
Figure A4b).

The correlations between the fold-changes in soluble forms of immune checkpoint proteins in plasma at Week 1 of TACE are shown in Figure 5. Again, the fold-changes in soluble checkpoint receptors and their ligands, including sCTLA-4 with sCD80 (*p* = 0.018; r = 0.50)/sCD86 (*p* < 0.001; r = 0.72) and sPD-1 with sPD-L1 (*p* < 0.001; r = 0.89), were positively correlated (Figure 5a). A strong correlation was noted between the fold-changes in the sCD80 and sTLR-2 levels (*p* < 0.001; r = 0.89), sCD40 and sHVEM levels (*p* < 0.001; r = 0.78), and sHVEM and sTIM-3 levels (*p* < 0.001; r = 0.72), respectively (Figure 5b).

## 3. Discussion

This study attempted the simultaneous quantification of 16 soluble immune checkpoint proteins in the plasma of patients with HCC who were observed during the early phase of treatment with lenvatinib or TACE. These immune checkpoint proteins include soluble forms of stimulatory or inhibitory factors, which modulate T-cell activation/proliferation and compose the cancer-immunity cycle [26]. In our previous report [21], we examined changes to these immune checkpoint proteins in sorafenib-treated HCC. The current results could offer additional data to compare changes in soluble checkpoint protein levels among tyrosine kinase inhibitors and TACE, which may clarify the immunomodulatory aspects of these treatments.

In this study, the plasma level of sCD27 was decreased and those of sCD40 and sTIM-3 were increased significantly at Week 1, while the level of sCD27 was increased significantly at Weeks 2 through 4 of lenvatinib treatment (Figure 1 and Figure 2). Although CD27 and CD40 are stimulatory factors and TIM-3 is an inhibitory factor, respectively, in the cancer-immunity cycle, soluble forms of stimulatory/inhibitory factors do not necessarily have set positive/negative immune effects; currently, the functions of these soluble proteins are yet to be fully defined. In short, (1) CD27 supports the antigen-specific expansion of naïve T-cells and is vital for the generation of T-cell memory [27]. A previous study found that circulating sCD27 constitutes a functional protein directly involved in CD8^+^ T-cell activation [28]. A persistent increase in the sCD27 level during Weeks 2 through 4 of lenvatinib treatment may reflect the CD8^+^ T-cell–related immunomodulatory effect of lenvatinib, which is in line with a previous study [29] revealing that the antitumor activity of lenvatinib was significantly diminished by CD8^+^ T-cell depletion. Further, (2) CD40 plays an important role mainly in the signaling pathways for the functioning of B-cells, monocytes, and dendritic cells [30]. In recent research [31], plasma sCD40 levels were upregulated in correlation with disease severity in patients with alcoholic hepatitis, indicating the existence of dysregulation of the immune system in chronic liver disease. Also, (3) TIM-3 promotes the exhaustion of T-cells in various types of cancer [32]. One study suggests circulating sTIM-3 might competitively bind to galectin-9 (a ligand of TIM-3), preventing a TIM-3/galectin-9–mediated immune response [33]. The increase in plasma sTIM-3 during lenvatinib treatment may restore immune exhaustion.

In vitro and in vivo preclinical studies have provoked the thoughts that there may be different immunomodulatory effects among similar tyrosine kinase inhibitors [34]. Lenvatinib may decrease tumor-associated macrophages, facilitate polarization from an M2-like phenotype toward an M1-like phenotype, and enhance CD4^+^ and CD8^+^ T-cell tumor infiltration, while sorafenib may have the opposite effects [35]. In our previous study of sorafenib [21], 11 of 16 soluble immune checkpoint proteins—most of which were inhibitory factors—experienced significant increases in plasma at two weeks of treatment. In this study, sCD27 displayed a significant change at two weeks. Both sorafenib and lenvatinib basically inhibit vascular endothelial growth factor receptors (VEGFRs) 1 through 3, fibroblast growth factor receptors (FGFRs) 1 through 4, platelet-derived growth factor receptor (PDGFR)-α, RET, and KIT. One potential explanation for the discrepancy is the existence of variable inhibitory profiles among the drugs against VEGFRs and FGFRs; specifically, lenvatinib is known to show more potent inhibitory activities than sorafenib against these receptor tyrosine kinases [36]. VEGFRs are particularly important because they activate various key components such as regulatory T-cells, tumor-associated macrophages, and myeloid-derived suppressor cells. Cytokines released by these cells inhibit natural killer cell activation and CD8^+^ T-cell proliferation, driving the emergence of an immunosuppressive microenvironment [37].

The tumor microenvironment actively participates in drug-resistance acquisition [38] in both primary and metastatic lesions [39] of HCC and other solid tumors [40]. Previous studies reported that an inflammatory microenvironment, circulating immune cells, and cytokines etc. play a significant role in the prognosis of HCC [41,42]. For example, the B-type Raf kinase (BRAF) mutation, one of the prognostic factors, could play a role in the response to tyrosine kinase inhibitors [43]. However, we found no significant difference in changes in soluble immune checkpoint protein levels between groups classified according to the response to sorafenib [21] or lenvatinib treatment, possibly due to the small sample size. Further investigation is required in a large group of patients to establish a determinant in driving clinical decision-making, which today are an unmet clinical need and a challenge for immunotherapy.

As part of our research, we also analyzed the levels of soluble immune checkpoint proteins in TACE-treated patients. Hepatic arterial embolization induces tumor necrosis and focal inflammation by blocking the tumor blood supply, which can impact cancer immunity by creating a source of tumor-associated antigens and enhancing the tumor-specific T-cell response [44]. Tampaki et al. [45] reported that TACE provokes a significant increase in the sTIM-3 level in plasma within the first week posttreatment, suggesting a reactive expansion of TIM-3 expression by T-cells as a negative feedback mechanism in response to intense immune stimulation following tumor necrosis. In our research encompassing a comprehensive measurement of 16 immune checkpoints, not only did the level of sTIM-3 but also those of many more soluble proteins changed significantly in the first week after TACE (Figure 3). However, some confounding factors may accompany TACE. When compared with systemic therapies, TACE presents a greater likelihood of inducing more sudden hypoxia in treated lesions and produces more hypoxia-related factors, which can influence components of cancer-immunity in the short-term. Therefore, the interpretation of results among TACE-treated patients is difficult and further research is necessary to better understand the treatment’s immunomodulatory effect.

Correlation analyses involving two soluble immune checkpoint proteins in plasma revealed that the fold-changes in soluble receptors and their ligands were positively correlated in both lenvatinib-treated and TACE-treated patients (Figure 4a and Figure 5a), suggesting that multiplexed fluorescent immunoassays are capable of accurate and simultaneous quantification of small amounts of the proteins. More interestingly, sCD40 at 1 week and sCD27 at two to four weeks, respectively, were positively correlated with sCTLA-4, sCD86, sPD-1, and sPD-L1 (Figure 4b and Appendix A
Figure A4b). While the mechanisms regulating these proteins remain unknown, sCD40 and sCD27 may be upregulated in cooperation with major checkpoint molecules such as sCTLA-4, sCD86, sPD-1, and sPD-L1. At Week 1 of TACE treatment, many more checkpoint pairs showed correlations unexpectedly (Figure 5b). For example, CD80 and TLR-2 have both been generally used as specific M1 macrophage surface markers in previous research [46]. Elsewhere, CD40 and HVEM messenger RNAs were coexpressed in the bioinformatic analysis of bladder cancer in the Cancer Genome Atlas database [47]. HVEM is a ligand of B- and T-lymphocyte attenuator (BTLA) and soluble BTLA in combination with sTIM-3 may be able to predict the rates of disease recurrence and survival among renal cell cancer patients [48].

Our study has several limitations. First, this is not a prospective study but instead a retrospective observational study. Moreover, randomization was not performed. In current guidelines [3,4,5], TACE is essentially indicated as a treatment option for patients with intermediate-stage HCC, while systemic therapy is suggested for those with advanced-stage HCC. Along these lines, in this study, the baseline characteristics of patients in the lenvatinib and TACE groups were not similar to one another (Table 1). It is therefore difficult to accurately compare the changes in soluble immune checkpoint levels in plasma between these treatment groups. Second, we could not collect plasma samples at Week 2 through 4 after TACE because of the less frequent hospital visits made by TACE-treated patients. We observed dynamic changes in soluble checkpoint levels in plasma at Week 1 of TACE but could not determine how long they persisted. Third, the sample size was small. Sample size affects the power or ability of all statistical tests to detect a relationship between two variables when it truly exists. The limitations of statistical analyses, such as the Wilcoxon signed-rank test and Spearman’s rank correlation test, are equally worth noting [49]. Lastly, we could not investigate associations between changes in soluble checkpoints and the outcomes of HCC patients because a majority of the included patients are still alive. Further studies are needed to clarify the clinical significance of circulating soluble checkpoint proteins in patients with unresectable HCC.

## 4. Materials and Methods

### 4.1. Patients

Between May 2018 and February 2020, we initiated lenvatinib treatment in 55 patients (46 males and nine females; median age: 73 (range: 42–85) years) with unresectable HCC at our institute. In this retrospective cohort study, 24 patients who received lenvatinib as a first-line systemic therapy (18 males and six females; median age: 75 (range: 55–89) years) were included. Lenvatinib therapy was principally indicated for patients with a good performance status and compensated liver disease for whom locoregional therapies were not indicated either due to the presence of vascular invasion, distant metastasis, or a TACE-refractory state, in accordance with available guidelines [3,4,5]. Lenvatinib (Lenvima^®^; Eisai Co. Ltd., Tokyo, Japan) was orally given to patients with unresectable HCC. The dose of lenvatinib was determined based on body weight, with initial dosages of 12 mg/day and 8 mg/day given to those over 60 kg and those under 60 kg, respectively. Plasma samples were collected at baseline (*n* = 24), during the first week (days 3–7 after initiating therapy; *n* = 19), and during the second to fourth weeks (days 8–28 after initiating therapy; *n* = 13).

As part of an effort to compare with lenvatinib treatment, we also studied 22 patients (15 males and seven females; median age: 76 (range: 44–86) years) who underwent conventional TACE during the same period. TACE was principally indicated for those with three or less HCC masses measuring greater than 3 cm or for four or more HCC masses in patients with Child–Pugh A/B hepatic reserve without extrahepatic metastasis or vascular invasion, in accordance with available guidelines [3,4,5]. Conventional TACE consisted of intra-arterial injection of lipiodol plus epirubicin followed by the injection of an embolic agent, Gelpart (Nippon Kayaku Co. Ltd., Tokyo, Japan), to interrupt blood flow. Plasma samples were collected at baseline and during the first week (days 3–7 after initiating therapy).

### 4.2. Soluble Immune Checkpoint Protein Assays

The plasma levels of 16 soluble immune checkpoint proteins were measured using multiplexed fluorescent bead-based immunoassays with the Milliplex Map Kit (EMD Millipore Corporation, Burlington, MA, USA) and the Luminex Bio-Plex-200 system (Bio-Rad Laboratories, Hercules, CA, USA): namely, sBTLA, sCD27, sCD28, sCD40, sCD80, sCD86, sCTLA-4, sGITR, sGITRL, sHVEM, sPD-1, sPD-L1, sTIM-3, sTLR-2, soluble lymphocyte-activation gene 3 (sLAG-3), and soluble inducible T-cell costimulator (sICOS). This kit enables simultaneous measurement of the concentration of these 16 checkpoint proteins in the plasma sample. In brief, the capture antibody-coupled beads were first incubated with antigen standards or samples for a specific time. The plate was washed to remove unbound materials, followed by incubation with biotinylated detection antibodies. After washing away the unbound biotinylated antibodies, the beads were incubated with a reporter streptavidin–phycoerythrin conjugate (SA–PE). Following the removal of the excess SA–PE, the beads were passed through the array reader, which measures the fluorescence of the bound SA–PE [50]. According to the manufacturer’s instructions, 12.5 μL of plasma was used for each measurement and all samples were assayed in duplicate; mean values were then adopted for further analysis. For values that were lower than the limit of detection, we used 10% of the lowest recorded value as a substitute [51].

### 4.3. Ethical Considerations

All patients supplied informed consent and the present study was conducted in accordance with the Declaration of Helsinki and was approved by the ethical committee of Osaka City University (#3719, approved on 12 July 2017).

### 4.4. Statistical Analysis

Analysis was conducted in Easy R (EZR) [52] (Saitama Medical Center, Jichi Medical University, Saitama, Japan) or R [53] (The R Foundation for Statistical Computing, Vienna, Austria) and figures were produced using the package ggplot2 [54]. Categorical variables were compared using Fisher’s exact test or the chi-squared test, when appropriate. Continuous variables were tested using the Mann–Whitney U test. Wilcoxon signed-rank tests were chosen to compare changes in soluble immune checkpoint concentrations during the early treatment period. Correlations of fold-changes in the levels of two proteins were determined with Spearman’s rank correlation test. A *p*-value of less than 0.05 was considered to be statistically significant.

## 5. Conclusions

The present study investigated changes in 16 soluble immune checkpoint proteins in the plasma of patients with HCC treated by lenvatinib or TACE. In the 24 lenvatinib-treated patients, the plasma level of sCD27 decreased but those of sCD40 and sTIM-3 increased significantly at Week 1, while the level of sCD27 was increased significantly at Weeks 2 to 4. These changes in soluble checkpoint protein levels during lenvatinib treatment were different from those seen during sorafenib treatment in our previous study [21], suggesting the two drugs present different inhibitory profiles against the receptor tyrosine kinases. Meanwhile, in the 22 TACE-treated patients, alongside the levels of sCD27, sCD40, and sTIM-3, those of sHVEM, sTLR-2, sCD80, sCTLA-4, sGITR, sGITRL, and sPD-L1 also increased. However, interpretation of the results among TACE-treated patients is difficult because TACE may be accompanied by some confounding factors, as discussed above. Further study is needed to better understand the immunomodulatory effect of the treatments, which may help future investigators to establish an effective combination immunotherapy.

## Figures and Tables

**Figure 1 cancers-12-02045-f001:**
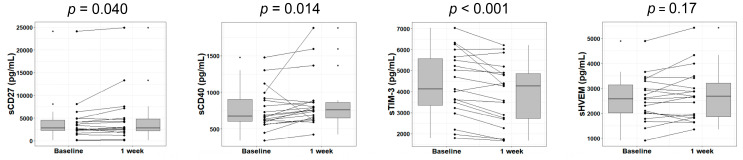
Changes in soluble immune checkpoint protein levels in plasma after 1 week of lenvatinib treatment. Box plots show the sCD27, sCD40, sTIM-3, and sHVEM levels in HCC patients at baseline and Week 1 of treatment. The vertical lengths of the boxes indicate the interquartile ranges and the lines in the boxes suggest the median values, while the error bars show the minimum and maximum values (in a range). The Wilcoxon signed-rank test was used. HCC, hepatocellular carcinoma; sTIM-3, soluble T-cell immunoglobulin and mucin domain–3; sHVEM, soluble herpesvirus entry mediator; sCD, soluble cluster of differentiation.

**Figure 2 cancers-12-02045-f002:**
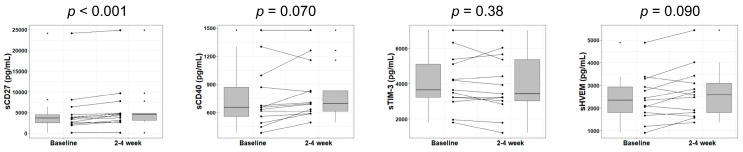
Changes in soluble immune checkpoint protein levels in plasma after 2 to 4 weeks of lenvatinib treatment. Box plots demonstrate the sCD27, sCD40, sTIM-3, and sHVEM levels in HCC patients at baseline and Weeks 2 through 4 of treatment. The vertical lengths of the boxes indicate the interquartile ranges and the lines in the boxes display the median values, while the error bars show the minimum and maximum values (in a range). The Wilcoxon signed-rank test was used.

**Figure 3 cancers-12-02045-f003:**
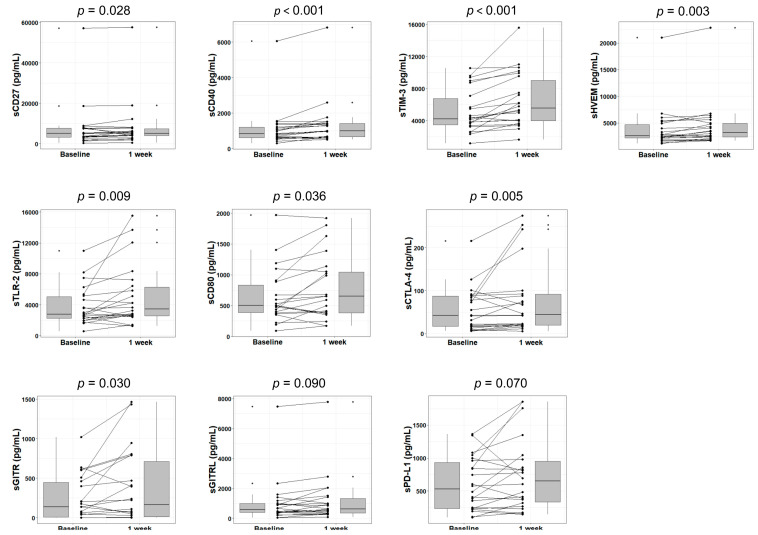
Changes in soluble immune checkpoint protein levels in plasma after 1 week of TACE treatment. Box plots display the sCD27, sCD40, sTIM-3, sHVEM, sTLR-2, sCD80, sCTLA-4, sGITR, sGITRL, and sPD-L1 levels in HCC patients at baseline and Week 1 of treatment. The vertical lengths of the boxes indicate the interquartile ranges and the lines in the boxes suggest the median values, while the error bars show the minimum and maximum values (range). The Wilcoxon signed-rank test was used. TACE, transarterial chemoembolization; sTLR-2, soluble Toll-like receptor 2; sCTLA-4, soluble cytotoxic T-lymphocyte antigen 4; sGITR, soluble glucocorticoid-induced tumor necrosis factor receptor; sGITRL, soluble glucocorticoid-induced TNFR-related ligand; sPD-L1, soluble programmed death-ligand 1.

**Figure 4 cancers-12-02045-f004:**
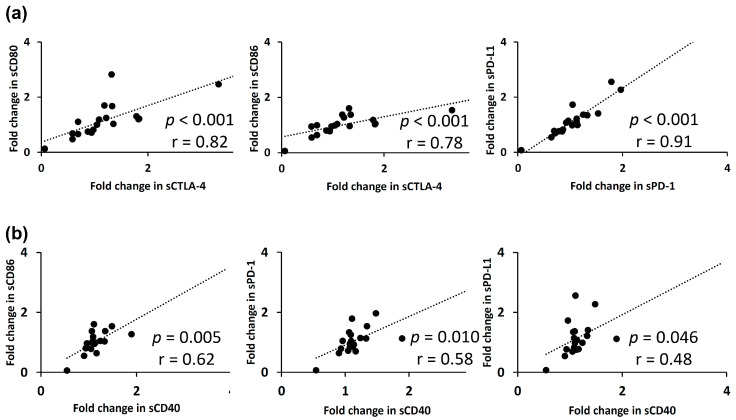
The relationships between fold-changes in soluble immune checkpoint protein levels in plasma at 1 week of lenvatinib treatment are presented. (**a**) A positive correlation was found between fold-changes in soluble checkpoint receptors and their ligands, including sCTLA-4 with sCD80/sCD86 and sPD-1 with sPD-L1, respectively. (**b**) sCD40 was positively correlated with some proteins in fold-changes, including sCD86, sPD-1, and sPD-L1. Spearman’s rank correlation test was used. A *p*-value of less than 0.05 was considered to be statistically significant.

**Figure 5 cancers-12-02045-f005:**
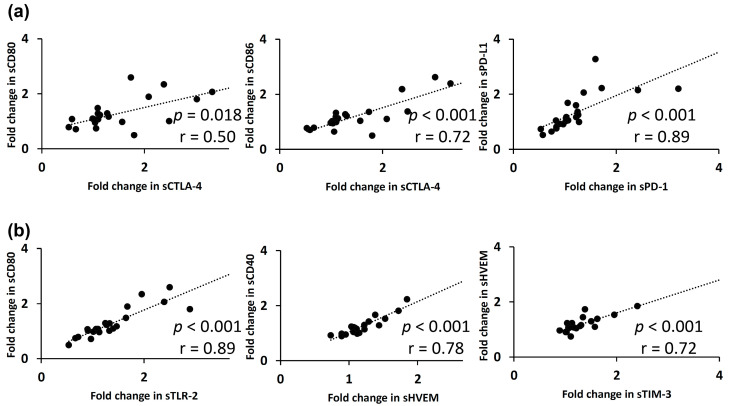
The relationships between fold-changes in soluble immune checkpoint protein levels in plasma at 1 week of receiving TACE are shown. (**a**) A positive correlation was found between the fold-changes of soluble checkpoint receptors and their ligands, including sCTLA-4 with sCD80/sCD86 and sPD-1 with sPD-L1. (**b**) Separately, a strong correlation was observed between the fold-changes in sCD80 and sTLR-2 levels, sCD40 and sHVEM levels, and sHVEM and sTIM-3 levels. Spearman’s rank correlation test was used. A *p*-value of less than 0.05 was considered to be statistically significant.

**Table 1 cancers-12-02045-t001:** Characteristics of hepatocellular carcinoma (HCC) patients at baseline.

Characteristics		Lenvatinib (*n* = 24)	TACE (*n* = 22)	*p* Value
Age	–	75 (69, 78)	76 (69, 80)	0.530
Sex	Male	18 (75.0)	15 (68.2)	0.746
	Female	6 (25.0)	7 (31.8)	
Etiology	Alcohol	5 (20.8)	3 (13.6)	0.536
	HBV	3 (12.5)	1 (4.5)	
	HCV	10 (41.7)	13 (59.1)	
	HBV + HCV	1 (4.2)	0 (0.0)	
	NASH	1 (4.2)	3 (13.6)	
	Unknown	4 (16.7)	2 (9.1)	
ECOG Perfomance Status	0 or 1	23 (95.8)	19 (86.4)	0.336
	2	1 (4.2)	3 (13.6)	
Aspartate aminotransferase		39 (27, 59)	39 (30, 50)	0.826
Alanine aminotransferase		27 (20, 52)	23 (16, 46)	0.567
Gamma-glutamyl transferase		62 (30, 122)	61 (32, 120)	0.930
Child-Pugh class	A	21 (87.5)	18 (81.8)	0.694
	B	3 (12.5)	4 (18.2)	
ALBI grade	1	8 (33.3)	8 (36.4)	0.999
	2	16 (66.7)	14 (63.6)	
α-Fetoprotein		59.4 (11.4, 1123.7)	21.2 (5.3, 55.9)	0.126
Des-γ-carboxy prothrombin		105 (72, 1312)	137 (56, 282)	0.605
BCLC stage	A	0 (0.0)	12 (54.5)	<0.001
	B	13 (54.2)	7 (31.8)	
	C	11 (45.8)	3 (13.6)	
Previous therapies	None	0 (0.0)	8 (36.4)	0.001
	Resection	5 (20.8)	2 (9.1)	0.418
	RFA/PEI	11 (45.8)	7 (31.8)	0.378
	TACE	23 (95.8)	9 (40.9)	<0.001
	HAIC	3 (12.5)	3 (13.6)	0.999
	Radiation	2 (8.3)	0 (0.0)	0.490
	Chemotherapy	0 (0.0)	2 (9.1)	0.223

Data are shown as median [interquartile range] or number (%). Abbreviations: HBV, hepatitis B virus; HCV, hepatitis C virus; NASH, non-alcoholic steatohepatitis; ECOG, Eastern Cooperative Oncology Group; ALBI, albumin–bilirubin; BCLC, Barcelona Clinic Liver Cancer; RFA, radiofrequency ablation; PEI, percutaneous ethanol injection; TACE, transarterial chemoembolization; HAIC, hepatic arterial infusion chemotherapy.

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
