# Peer review of "Early Change in the Plasma Levels of Circulating Soluble Immune Checkpoint Proteins in Patients with Unresectable Hepatocellular Carcinoma Treated by Lenvatinib or Transcatheter Arterial Chemoembolization"

_cancers, 2020, doi:10.3390/cancers12082045_

Round 1

Reviewer 1 Report

Thank you for the opportunity to review this manuscript.
The topic looks interesting. There are problems, however, in the manuscript that needs the authors' attention:

Abstract- line 26 should be "limited."
Abstract - line 27, please add "several centres."
Introduction- line 32 Ref 1- Check the Cancer Statistics-2020 Ref
Introduction- lines 57/58 State your research question and hypothesis- you could also add what we know and what we do not know and how this study can help.
Table 1- "Others" - State them and the number for each.
_Table 1- State the full name of each abbreviation. Keep under the table for notes to explain things.
-Discussion, discuss your findings against those of
- Kawamura Y et al., 2020.
- Kanada SR et al. 2020
- Liu Z et al., 2019.
- Cheng H et al., 2019
-Limitations 243-Also add no randomisation in allocation patients to each group.
- Material - 287 What were the inclusion and exclusion criteria? How were the patients allocated to each group, or this was a retrospective collection. In this case, were there any reasons for the selection of procedure by clinicians.

-Line 290- State number of approval and date/year.
-Line 292-Add full name for words such as EZR.
-References - update and add suggestions given.

Reviewer 2 Report

The authors provide a nice analysis on possible soluble checkpoint proteins (IC) pointing toward a potential immunomodulatory effect between anti-angiogenic agents and TACE in HCC. Although, as correctly reported by the authors, the small sample size, the retrospective nature and the lack of a validation cohort are limitations to be considered, this study could provide a rationale for further studies. Nonetheless, major issues should be addressed.

For this reason, I would suggest changing conclusions highlighting that the results presented are hypothesis generating rather than conclusive.

  1. Cytokine, checkpoint proteins and angiogenic factors seem to dissect some immunomodulatory effects between anti-angiogenic agents and TACE in HCC authors cohort.  Did the authors confirm by ELISA those results?

In more detail, ELISA kit in order to quantify the level of the variable examined it is a worthy tool (either by DuoSet or Quantikine based kit and); importantly, it is not clear if the authors generated data supporting the precision of the assay. In other words, sensitivity, precision (repeatability), accuracy, matrix effects, and selectivity of this tools are of paramount relevance in evaluating the accountability of the proposed methods.

  1. The hypothesis evaluated with the Wilcoxon matched pairs signed ranks test is whether or not the median of the difference scores equals zero. Let us consider the situation of x measurements tending to exceed y measurements in the low range and vice versa in the high range, with similar values in the mid-range. Such results may have a median of the difference scores of approximately zero; that is, there might be no significant differences by the Wilcoxon matched pairs signed ranks test, although there would be differences by linear regression (Deming or Passing-Bablok) and/or difference plots lacks both a regression equation (proportional and constant error) and difference plots. Despite probably beyond the scope of this manuscript, I would be interested in the authors comment to this issue and I would suggest to highlight some of this concept within limitations discussion of this manuscript.

  1. Sample size affects the power or ability of all statistical tests to detect a relationship between two variables when it truly exists. Correlation coefficient is no different and can give a false negative result if a sample size is inadequate. Hence, the smaller the correlation coefficient between two variables the investigators would like to detect in a study, the larger the sample size is required. Because Spearman’s correlation coefficient is not as efficient as Pearson’s coefficient, an extra 10% in sample size is needed to achieve the same statistical power if Spearman’s correlation coefficient is the targeted outcome instead of Pearson’s correlation coefficient. Again, can the author comment on this (see point 2.)

  1. General comment on introduction/discussion: some of the manuscript-mentioned annotation can significantly improve the manuscript quality and provide important information for the scientific community. IC seem to represent a major pathway involved in the gene expression signature. In the discussion section: I think it is important to mention that particular study refers to these immune-related pathways, the authors correctly mentioned reff. 23, 33. Nonetheless, the related mechanisms within this particular HCC and other gastro-intestinal tumors model: indeed, a tight correlation exists between, immune-infiltrate, angiogenesis and cancer progression, and dissemination to distant sites and to nodal compartment. Indeed, immune cells come and go across the permeable capillaries. Because of these intimate interactions, the capacity of dendritic cells and endothelial cells ECs as antigen-presenting cells (APC) can be also discussed, since several examples have been recently published and can pioneer future ICI unexplored application (i.e. PMID: 31627433; PMID: 31277479), especially while explaining the proangiogenic neoplastic phenotype. Moreover, while discussing sorafenib and TKI effect on the described HCC phenotypes, the authors state: "lenvatinib may decrease tumor-associated macrophages, facilitate polarization from an M2-like phenotype toward an M1-like phenotype, and enhance CD4+ and CD8+ T-cell tumor infiltration, while sorafenib may have the opposite effects".

In the frame of this thinking, I personally miss some important insights matching predicting factors for sorafenib response and an inflammatory micro-environment and circulating immune cells and cytokines play a significant role in HCC prognosis (PMID: 31640191). Indeed, A great number of data showed that the limited clinical success of these drugs is probably due to the complex relationship between cancer cells and tumor microenvironment in HCC. Therefore, halting The secretion of chemokines is also influenced by different doses of MEK and BRAF inhibitors. They inhibit the secretion of interleukin-8 (IL-8) and VEGF and exert an influence on tumor microenvironment with the arrest of proangiogenic activities (tumor escape concept) and the development of metastases.

Collectively,  I was wondering if the authors could develop a bit more the general concept that seems to indicate that immunemicroenvironment are characterized by a peculiar immunophenotype, determinant in driving clinical decision also in neoplasia that are today an unmet clinical need and a challenge for immunotherapy (as HCC is, also in unusual metastatic sites). Maybe some papers have been published about this phenomenon (i.e. PMID: 32064051)?

Moreover, it is well known that TKI, and specifically BRAF inhibition, play a pivotal role in impacting HCC outcome (PMID: 31640191). Since both sorafenib and lenvatinib inhibit vascular endothelial growth factor receptors (VEGFRs) 1 trough 3, fibroblast growth factor receptors (FGFRs) 1 through 4, platelet-derived growth factor receptor (PDGFR)-α, RET, and KIT IC (orchestrate relevant downstream signaling in HCC biology, it is tempting to speculate a potential combination of BRAF-directed therapy with (PMID: 31766556) while envisioning a rational for future target therapy pioneering a synergy between BRAF-targeted therapy and immunotherapy with enhanced survival and T-cell function.    

The manuscript would benefit from a native-speaker revision and figure beautification.

Round 2

Reviewer 2 Report

The authors have clarified several of the questions I raised in my previous review. Most of the major problems have been addressed by this revision. No further comments from this reviewer.